# Clonal Myeloid Dysplasia Following CAR T-Cell Therapy: Chicken or the Egg?

**DOI:** 10.3390/cancers15133471

**Published:** 2023-07-03

**Authors:** Vladimir Vainstein, Batia Avni, Sigal Grisariu, Shlomit Kfir-Erenfeld, Nathalie Asherie, Boaz Nachmias, Shlomtzion Auman, Revital Saban, Eran Zimran, Miri Assayag, Kalman Filanovsky, Netanel A. Horowitz, Eyal Lebel, Adir Shaulov, Michal Gur, Chaggai Rosenbluh, Svetlana Krichevsky, Polina Stepensky, Moshe E. Gatt

**Affiliations:** 1Department of Hematology, Hadassah Medical Center, Faculty of Medicine, Hebrew University of Jerusalem, Jerusalem 91121, Israelaumann@hadassah.org.il (S.A.); rmoshg@hadassah.org.il (M.E.G.); 2Department of Bone Marrow Transplantation and Cancer Immunotherapy, Hadassah Medical Center, Faculty of Medicine, Hebrew University of Jerusalem, Jerusalem 91121, Israel; 3Department of Hematology, Kaplan Medical Center, Rehovot 76100, Israel; 4Faculty of Medicine, Hebrew University of Jerusalem, Jerusalem 91121, Israel; 5Department of Hematology, Rambam Medical Center, Faculty of Medicine, Technion University, Haifa 32000, Israel; 6Department of Human Genetics, Hadassah Medical Center, Faculty of Medicine, Hebrew University of Jerusalem, Jerusalem 91121, Israel

**Keywords:** multiple myeloma, myelodysplastic syndrome, CART therapy

## Abstract

**Simple Summary:**

Patients suffering from multiple myeloma receive many lines of treatment including chimeric antigen receptor T cell (CART) therapy. Myeloma patients may develop devastating secondary malignancies such as myelodysplastic syndrome (MDS). The causative relationship between various anti-myeloma treatments and MDS is not fully understood. We report a study of five CART-treated patients myeloma patients who had MDS (four only after CART and one already prior to CART). We found that all these patients had all their MDS-related molecular changes (mutations) already prior to CART even if they did not have overt MDS in bone marrow evaluation before CART. No new mutations developed after CART, but the frequency of pre-existing mutations did increase. Our study presents evidence that anti-myeloma CART therapy may promote the expansion of pre-existing MDS clones rather than causing the development of new ones.

**Abstract:**

Multiple myeloma (MM) is characterized by recurrent relapses. Consequently, patients receive multiple therapy lines, including alkylating agents and immune modulators, which have been associated with secondary malignancies such as myelodysplastic syndrome (MDS). Anti-B-cell maturation antigen (BCMA) chimeric antigen receptor T cell (CART) therapy is efficacious in patients with relapsed/refractory (R/R) MM. However, the long-term complications, particularly MDS, are not well understood. Whether CART therapy causes or promotes MDS has not been thoroughly investigated. In this study, we explored the causal relationship between MDS and CART therapy. We retrospectively examined the prevalence of MDS-related morphological and mutational changes before and after administration of CART therapy in five patients. Among them, four developed MDS after CART therapy, while one had pre-existing MDS prior to CART. None of the four patients who developed post-CART MDS showed morphological MDS changes prior to CART therapy. However, all four patients exhibited molecular alterations associated with MDS in their pre-CART as well as post-CART therapy bone marrow. No new mutations were observed. Our findings provide initial evidence suggesting that anti-BCMA CART therapy in MM may promote expansion of pre-existing MDS clones rather than causing development of new clones.

## 1. Introduction

Advances in therapy of multiple myeloma (MM) including novel agents such as proteasome inhibitors (PIs) and immunomodulators (IMiDs) as well as anti CD38 antibodies and other advanced therapies greatly improved patient outcomes with profound prolongation of survival [1,2]. This prolonged survival, however, is associated with the risk of developing concurrent or secondary primary malignancies (SPM), the most devastating and treatment-limiting of which are the myeloid malignancies, myelodysplastic syndrome (MDS) and acute myeloid leukemia (AML). It has been reported that MM as well as monoclonal gammopathy of unknown significance (MGUS) patients have 8–11 times increased risk to develop MDS/AML due to both therapy-related and therapy-unrelated factors [3,4,5,6,7,8]. Presence of preleukemic mutations without overt MDS/AML (clonal hematopoiesis or CHIP) is not uncommon in MM patients but it is not necessarily associated with an increased risk of subsequent myeloid malignancies [9,10]. Importantly, a recent publication demonstrated the ability of lenalidomide, one of the most widely used anti-myeloma drugs, to promote TP-53-mutated myeloid neoplasms by giving selective growth advantage to the pre-existing TP53-mutated hematopoietic stem cell clones [11]. Given the elevated intrinsic and extrinsic overall risk of MDS/AML in MM population, it is reasonable to assume that the cumulative risk of AML/MDS development will increase as these patients live longer due to advances in anti-myeloma therapies and supportive care. 

Despite major recent advances, the vast majority of MM patients develop resistance to all currently available treatments. In relapsed/refractory (RR) MM [2], excellent responses have been reported to B-cell maturation antigen (BCMA)-targeting immunotherapy, and specifically to anti-BCMA chimeric antigen receptor T-cell (CART) therapy [9,12,13,14,15]. Idecabtagene vicleucel (Ide-cel) and Ciltacabtagene autoleucel (Cilta-cel) are the two recently FDA-and EMA-approved anti-BCMA CART-based therapies that induce deep responses (over 85% ORR rates), in heavily pretreated patients [16,17]. However, while the short-term complications of CART therapy are known and manageable, long-term adverse events are not well-defined and the data are emerging [18,19]. The observation of high frequency of pre-leukemic mutations and MDS/AML after CART (Andersen MJ, ASH 2022 abstract #4935) raises the question whether CART therapy by itself increases the frequency of the myeloid mutations (similarly to lenalidomide [11]). Alternatively, it might indicate an increased risk of MDS/AML in a heavily pretreated population, or both mechanisms may contribute to the development of leukemogenesis. Herein, we set to show that CART by itself does not necessarily cause the clonal myeloid disorders, but it may actively promote expansion of the pre-existing mutated clones. 

Recently, we published the short-term safety and initial efficacy [20,21] of a novel academic anti-BCMA CAR construct (namely, HBI0101), in a phase I clinical trial for the treatment of R/R MM patients (NCT04720313). In long-term follow-up, we observed several cases of post therapy MDS/AML. In order to determine the causal relationship between CART therapy and post-CART MDS, we performed a detailed analysis of the morphologic features, somatic mutations, and cytogenetic changes in patients treated within the HBI0101 trial who developed clinically significant clonal myeloid disorder (i.e., MDS/AML).

## 2. Materials and Methods

### 2.1. BCMA-CAR (HBI0101) Clinical Trial

The phase Ia clinical trial was previously reported [20,21]. This was a single-center phase I clinical trial exploring the safety and efficacy of the HBI0101 in-house manufactured anti-BCMA CART cell product. Briefly, we enrolled R/R MM and amyloidosis patients after at least three prior lines of treatment including a proteasome inhibitor, an immuno-modulator (IMiD), and an anti-CD38 antibody. The first part of the trial consisted of the administration of HBI0101-transduced T cells, at escalating cell doses of 150-, 450-, and 800 × 10^6^ CAR+ cells. The lymphodepleting protocol consisted of fludarabine 25 mg/m^2^ and cyclophosphamide 250 mg/m^2^ given on days −5 to −3 before CAR-T cell infusion. The complete study protocol, eligibility criteria, study design, initial safety, and efficacy results were published in the aforementioned report [20,21]. 

### 2.2. Patient Clinical Data Assessment

Patient data and samples including bone marrow (BM) aspirates were collected and processed according to the institutional protocol. Pathological evaluation, immunophenotyping, and cytogenetic analysis were performed using a predetermined follow-up schedule. Furthermore, mutation analysis of patients with suspected MDS/AML was carried out using Archer^®^ VariantPlex^®^ myeloid panel (see Appendix A for the full list of the genes included in the mutational analysis). Semi-quantitative P53 and quantitative RUNX1 assessments were done as previously described [22]. Allele specific PCR DNA samples from three time points were tested for patient specific RUNX1 mutation. Mutant and wild type allele quantification was performed by allele-specific Real Time PCR (StepOne Plus, Applied Biosystems Waltham, MA, USA) and allelic ratio calculated with the comparative delta Ct(∆∆Ct) PCR method (https://www.gene-quantification.de/chapter-3-pfaffl.pdf (accessed on 15 March 2023)). The individual patient-specific PCR system designed for RUNX1 c.954_958del p.Ser319ThrfsTer279 mutation includes: allele-specific reverse primers for mutant allele and common forward primer and Taqman probe for both mutant and wild type alleles. The amplification of mutant and wild type alleles was performed in separate reactions. Two different sets of primers were tested to obtain the best discrimination between mutant and wild type alleles and avoid false positive results.

## 3. Results

### 3.1. Characteristics of the MM Patient Treated with HBI0101 Developing Myeloid Disorders 

The phase I clinical trial of anti-BCMA CART in R/R MM patients has been previously described [20,21]. Among the 40 patients treated to date with a median follow-up of 6 months, four patients developed overt post CART MDS and are reported herein. In addition, we report a fifth patient who received compassionate CART therapy for progressive amyloidosis diagnosed with therapy-related MDS prior to CART treatment. MDS was diagnosed in bone marrow aspirates and/or biopsies, which were performed as evaluation of prolonged cytopenias. Morphological features were typical of MDS and included hypercellularity, trilineage dysplasia, and left shift in all patients. No ringed sideroblasts were found. Patients’ baseline characteristics, history, and prior lines of treatment are summarized in Table 1. All patients were refractory to their last treatment line and exposed to alkylating agents such as high dose melphalan, low dose melphalan, and/or cyclophosphamide, while three were penta-refractory. The patients were treated for prolonged periods of time (3.5–11.5 years) with 1–3 types of immunomodulatory drugs (IMiDs). Prior to the CART therapy, four of these patients had mild to moderate uni- or bi-cytopenia (with mean corpuscular volume (MCV) within normal range), which was attributed either to MM disease involvement or to toxicity of their previous treatment. Prior to CART therapy, none of the four patients showed morphological dysplastic features of their bone marrow, which, as expected in heavily pre-treated patients, was hypocellular.

### 3.2. Toxicity and Efficacy of the CART Therapy

All patients experienced significant cytopenia after CART therapy (Table 2). Partial or complete recovery of counts was seen in patients 1 to 4. In patient 5 with the pre-existing MDS, neutropenia and anemia recovered to grade 2, while grade 4 thrombocytopenia persisted, and repeated biopsy of bone marrow after 30 days of treatment showed severe hypocellularity. 

All five patients achieved complete remission of their underlying disease, showing minimal residual disease (MRD) negativity (10^−5^) as evaluated by flow cytometry. 

### 3.3. Clonal Myeloid Neoplasm Characteristics and Outcomes

Patients 1–4 have achieved partial or complete blood counts recovery. However, at different time points during the follow-up, 135–360 days post CART therapy (Table 3), there was a recurring decline in the blood counts accompanied by an increase in red blood cell MCV, raising suspicion of MDS. Subsequent bone marrow aspiration and biopsy for all four patients revealed morphologic dysplastic changes with an elevated blast count, and cytogenetic as well as molecular alterations characteristic of MDS (Table 3). It is noteworthy that at the time of MDS assessment and diagnosis all patients had negative MRD for MM (aberrant PC clones of less than 10^−5^). Among the remaining 35 patients enrolled in the trial, none had experienced a recurring decline in blood counts suggestive of MDS or AML. Moreover, their bone marrow assessments, conducted as a part of the study, did not demonstrate morphological evidence of MDS.

Although 4 out of 5 patients had high or very high risk for progression to AML as evident by the revised international prognostic scoring system (R-IPSS) [23], and molecular changes calculated IPSS-M [24,25], none developed AML, including patient 5 who had pre-CART MDS. However, it should be emphasized that the duration of follow-up since MDS diagnosis was short (2–12 months) due to death of 4 out of 5 patients.

Patient 1 died of COVID-19 complications in ongoing CR and MRD negativity. Patient 2 developed MDS with 15% blasts, yet eventually died due to aggressive extra-medullary MM relapse. Patient 3 succumbed to MDS-related complications while in ongoing MM remission (he received one course of 5-azacytidine with severe neutropenia and declined further MDS directed therapy). Patient 4 is in ongoing MM remission and has a stable MDS clone with low risk MDS and requires no treatment. Patient 5 had a short duration of response and died due to aggressive relapsed MM, without evolution of his previous MDS clone (i.e., no increase in blast percentage and no appearance of new molecular alterations, compared to his pre-CART therapy status).

### 3.4. Retrospective Analysis of Pre-MDS Features

As shown in Table 4, post-CART therapy BM samples of all five patients demonstrated MDS-related mutations. Bone marrow samples obtained prior to CART therapy were revised, and no morphological MDS features were evident (excluding patient 5). However, we found the same MDS-related mutations as in post-CART samples, albeit at lower allelic frequency, with no appearance of new molecular abnormalities after CART therapy. Karyotype analysis was not conducted on the pre-CART bone marrow samples of patients 1–4 since it is not a routine assessment in the study. For patient 1, pre-CART bone marrow was available for fluorescence in situ hybridization (FISH) analysis showing the presence of 1 out of 20 cells with deletion of chromosome 7.

## 4. Discussion

A significant proportion of patients with MM may develop myeloid malignancies over the course of disease. This has been previously reported and attributed to chemotherapy exposure (particularly melphalan) as well as the prolonged usage of IMIDs [3,4,5,6,7,8,10,11]. Recent reports imply variable rates of MDS following CAR T-cell therapy [12,17,18,19,26,27,28].

In the published CART clinical trial of Ciltacabtagene Autoleucel, 10% of heavily pretreated patients developed MDS or AML after a median follow up of 28 months [28]. The causative role of CAR T-cell therapy in the development of myeloid disorders is still not fully understood.

Recent reports [ASH 2022 #4935] found the presence of mutations associated with myeloid neoplasms in one-quarter of the patients who had next generation sequencing performed on DNA samples after CART therapy for lymphoma and MM. Three patients with post-CART MDS (two treated for lymphoma and one for myeloma) were found to have had cytogenetic changes prior to CART therapy [18,28]. Additionally, six patients were reported with post-CART MDS/AML for whom paired pre- and post-CART mutational panels were done, and in 4 out of 6 patients MDS/AML-related changes were already seen prior to CART (all patients treated for lymphoma, all pre-CART samples collected from peripheral blood rather than bone marrow) [29,30]. Notably, there is no available information regarding the evaluation for MDS-related morphological and histological changes in pre-CART samples of these patients with post-CART MDS [20,28,29,30]. Therefore, it cannot be established that MDS had not been present prior to the administration of CART therapy.

In our series of 40 heavily pretreated MM patients receiving anti-BCMA CART, after a median follow up of 6 months, four patients developed MDS consistent with the incidence reported in the Ciltacabtagene Autoleucel cohort [28]. Therefore, our objective was to investigate the mutational burden of these patients before and after receiving CART therapy. None of our patients exhibited morphologic dysplastic changes in their pre- CART bone marrow samples. Although some patients experienced cytopenias prior to CART therapy, these were attributed to the last treatment or MM disease burden. Evidence of clinical MDS, characterized by worsening cytopenias, an increase in MCV, and the presence of morphologic dysplasia in bone marrow aspirates, was observed only following CART therapy. Upon retrospectively analyzing the MDS-related mutational alterations, no new mutations were detected after CART therapy. Instead, all mutations identified post-CART were already present before therapy, albeit at significantly lower allelic frequency. Since all four patients achieved MRD negative remissions of MM after CART therapy, none of these mutations can be attributed to the malignant plasma cell clone. Furthermore, there was an increase in mutational load following CART therapy, consistent with the emergence of clinically significant MDS, which was previously occult prior to the CART therapy.

The mechanism of the progressive growth of the pre-existent malignant myeloid clones after CART therapy requires further investigation. It is possible that the observed profound immune suppression following CART therapy contributes to the expansion of myeloid disease clones. In line with previous reports [28], in our patients, overt MDS developed within 135–360 days. While the immune suppression following CART therapy is prolonged, it is expected to wane over time due to the limited longevity of the CAR engineered T cells. Another hypothesis suggests that the presence of malignant plasma cells in the pre-CART bone marrow may suppress the expansion of the myeloid MDS clones. Once the malignant PC clone is eliminated by the CART treatment, the myeloid clone can emerge.

During CART therapy, there is a well-known occurrence of profound myelosuppression, lymphodepletion, and immunosuppression. Consequently, the suppression of immune surveillance as well as relative proliferative advantage of pre-existent MDS clones over the normal bone marrow stem cells could allow clinically evident expansion of pre-existent MDS clones. Since MDS post CART therapy has been also reported in lymphoma trials, this may reflect a pan-CART therapy property [18].

Given the high prevalence of clonal hematopoietic changes [9,10] in heavily pre-treated MM patients, we believe that analyzing the mutational status prior to CART therapy will not alter treatment strategy due to the limited therapeutic options in these highly refractory patients. As clinical development of CART therapy involves its use in earlier therapy lines, it is prudent to better understand and track the origins of the MDS myeloid clones. Considering these findings, it is crucial to evaluate the risk of developing MDS for all the different treatment strategies in MM, in order to personalize treatment choices.

Sequencing of pre- and post-CART bone marrow samples of all enrolled patients was not included in the trial and is beyond the scope of this report. Previous reports systematically evaluated the presence of myeloid malignancy-related mutations either prior or after CART, but not in the paired fashion. Conducting such an analysis in future trials could provide valuable insight into the incidence of pre-CART malignant myeloid clones and the clinical and biological features of patients in whom these clones progress into overt MDS/AML rather than remaining subclinical.

## 5. Conclusions

In conclusion, our study is the first to investigate pre- and post-CART therapy bone marrow samples in myeloma patients with post-CART MDS. Our findings suggest that development of myeloid malignancies after CART therapy for MM may be attributed to the expansion of pre-existing malignant myeloid clones, rather than their de novo development. We propose several potential mechanisms for this phenomenon, which warrant further in vitro investigation and within the context of CART clinical trials.

## Figures and Tables

**Table 1 cancers-15-03471-t001:** Baseline patients’ characteristics prior to CART.

	Patient 1	Patient 2	Patient 3	Patient 4	Patient 5
Age	64	52	72	70	63
Gender	m	m	m	m	m
MDS prior to CART	No	No	No	No	Yes
MM ISS	1	1	2	2	1
Plasma cells in BM (%)	3	20–30	50	50	15
MM-related FISH	Normal	Trisomy 11, T (14:16)	1q amplification	Hyperdiploid	T (11:14)
Leukopenia Anemia Thrombocytopenia	No No No	No Yes Yes	No Yes Yes	Yes Yes Yes	Yes Yes Yes
LDH (u/L, ULN = 246)	220	190	180	240	275
MCV	95	94	92	84	107
BM cellularity	Normal	Low	Low	Low	High
Morphologic dysplasia in BM	None	None	None	None	Yes
Blasts % in BM	<1	<1	<1	<1	3
Prior lines of anti-myeloma therapy	8	7	4	6	10 *
Best response/ which line	VGPR/3rd	VGPR/1st–2nd	CR/1st	VGPR/1st	CR/1st and 4th
Previous ASCT ** (# years before CART)	Yes (10)	Twice (8, 11)	Yes (3)	Twice (0.7, 10)	Yes (3.5)
Use of chemotherapy in MM therapy prior to CART (not including lymphodepletion)	Cyclophosphamide Melphalan	Cyclophosphamide Melphalan Cisplatin Adriamycin etoposide	Cyclophosphamide	Cyclophosphamide	Cylophosphamide
Previous IMiD	Thalidomide Lenalidomide Pomalidomide	Thalidomide Lenalidomide	Lenalidomide Pomalidomide	Thalidomide Lenalidomide Pomalidomide	Lenalidomide Pomalidomide
Total Duration of therapy with IMiDs (years)	9.5	4.5	3	7	2.5
Years since MM diagnosis	10.5	11.5	3.5	9	4.5
Triple refractory	Yes	Yes	Yes	Yes	Yes
Penta refractory	Yes	Yes	No	No	Yes
Other agents exposed	Venetoclax	Belantamab Selinexor	Belantamab		Belantamab Venetoclax Selinexor

Abbreviations: ASCT—autologous stem cell transplantation; BM—bone marrow; CR—complete response; IMID—immunomodulatory drug; ISS—international scoring system; LDH—lactate dehydrogenase; MCV—mean corpuscular volume; ULN—upper limit of normal; VGPR—very good partial response * In addition to MM, MDS was treated with 5-azacytidine for 5 courses prior to CART. ** Conditioning with high dose melphalan in all patients.

**Table 2 cancers-15-03471-t002:** CART administration, toxicity, and efficacy.

	Patient 1	Patient 2	Patient 3	Patient 4	Patient 5
CAR+ cells infused (×10^6^)	150	450	800	800	450
Adverse events of interest					
CRS grade	None	2	1	2	3
CRS duration (days)	0	2	0	1	2
Hematologic adverse events					
Neutropenia (grade)	Yes (1)	Yes (4)	Yes (4)	Yes (3)	Yes (4) *
Anemia (grade)	No	Yes (2)	Yes (2)	Yes (3)	Yes (3)
Thrombocytopenia (grade)	No	Yes (4)	Yes (3)	Yes (4)	Yes (4) *
Duration of Hematologic toxicity ** (days)	4	90	144	125	NA
Febrile Neutropenia	No	Yes	Yes	No	Yes
Efficacy					
Best hematologic response	CR	CR	CR	CR	CR
MRD negativity Day 30 Day 180	Yes Yes	Yes No	Yes Yes	Yes Yes	Yes No
Time to best confirmed response (days)	30	30	60	30	30
DOR (months)	10	6	17	17 Ongoing	4

Abbreviations: CR: complete remission; CRS: cytokine release syndrome; DOR: duration of response; MRD: minimal residual disease; * Started with grade 4 due to MDS. ** Duration to resolution to grade 2 or better.

**Table 3 cancers-15-03471-t003:** Myeloid neoplasm features.

	Patient 1	Patient 2	Patient 3	Patient 4	Patient 5
Time from CART to MDS diagnosis (months)	9	6	5	13	NA
Progression to AML	No	Yes	No	No	No
R-IPSS	Very High (7)	Very high (7)	Intermediate (3.5)	Very low (1.5)	Very High (6.5)
IPSS-M *	Very high (2.54)	Very high (2.75)	High (1.39)	Very low (−1.17)	Very high (1.86)
Cytopenias: WBC Hemoglobin MCV Platelets	2.5 8 79 43,000	2.5 9.1 108 33,000	2.9 8.9 105 53,000	3.7 12.8 100 76,000	2 8.8 110 14,000
Blasts (%)	5	15	0.7	0	2
Cytogenetic abnormalities	44, XY, −3, der (5), −7 [1]/46, XY, −3, der (5), +mar1 [1]/43, idem, −21 [3]/44–45, idem, +mar1, +mar2, +mar3 [5]	46, XY, +1, der (1;7) (q10;p10) [10]	46, XY, −5, +mar [7]/46, XY, del (5) (q13q34) [2]/46, XY [11]	46, XY [20]	45, XY, −7 [4/30]
Mutated genes	TP53	RUNX1, DNMT3A, PHF6, PPM1D, RUNX1	TP53 (2 different mutations)	RAD21, TET2	RUNX1, DNMT3A, TET2
Survival since MDS diagnosis (months)	1	2	12	4 Ongoing	7
Survival (months) from CART/Cause of death	10 COVID-19 infection	8 MM	17 Infection	17 Alive	7 MM

IPSS-R—revised international prognostic scoring system; IPSS-M—molecular international prognostic scoring system * https://mds-risk-model.com/ (accessed on 1 February 2023).

**Table 4 cancers-15-03471-t004:** Retrospective bone marrow analysis prior to CART.

	Patient 1	Patient 2	Patient 3	Patient 4	Patient 5
Dysplastic morphologic abnormalities prior to CART	No	No	No	No	Yes
Presence of Cytogenetic abnormalities before CART	Yes (1/200 45XY − 7)	Unknown	Unknown	None	Yes (26/200 45XY − 7)
Presence of Molecular Changes before CART	Yes	Yes	Yes	Yes	Yes
Change in VAF (% before and after CART)					
TP53	15% to 89%		2% to 10%		
RUNX1		20% to 44.8%			3.9% to 1.4% *
DNMT3A		8% to 0.5%			
PHF6		41% to 86%			
CUX1		26% to 82%			
RAD21				1.5% to 10.5%	
TET2				3.5% to 6.5%	

Abbreviations: VAF variant allelic frequency. * In patient 5, out of three mutations, only RUNX1 mutation was quantitatively compared prior and post CART.

## Data Availability

Individual data are unavailable due to privacy and ethical restrictions.

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
