# Peer review of "Clonal Myeloid Dysplasia Following CAR T-Cell Therapy: Chicken or the Egg?"

_cancers, 2023, doi:10.3390/cancers15133471_

Round 1
Reviewer 1 Report
The clinical data as well as the morphological, cytogenetic and molecular studies carried out in 5 patients with advanced MM before and after treatment with CART are, in my opinion, very relevant since they show how, after these therapies, patients with genetic profiles and subclinical molecular alterations coherent with the existence of myelodysplastic clones perhaps related to MM or previous treatments, they can develop advanced myelodysplasias after CART therapies. The evidence is very well documented, the arguments adequately expressed as well as the bibliographic context.
Author Response
We appreciate very much the reviewer's positive notes. Our understanding is that there are no comments to address.
Reviewer 2 Report
General comment:
Authors retrospectively examined the prevalence of MDS-related morphological and mutational changes before and after administration of CART therapy in five patients. Among the 40 patients treated to date, four patients developed overt post CART MDS and in addition, they report a fifth patient who received CART therapy for progressive amyloidosis diagnosed with therapy-related MDS prior to CART treatment. None of the four patients who developed post-CART MDS showed morphological MDS changes prior to CART therapy. Furthermore, upon retrospectively analyzing of MDS-related mutational alterations, no new mutations were detected after CART therapy. These findings provide initial evidence suggesting that anti-BCMA CART therapy in MM may promote expansion of pre-existing MDS clones rather than causing development of new clones. This report is considered as important information for physicians. If the authors could add information as below, this paper would be more informative.
Specific comment:
1. It would be helpful if authors could add the information on the genes tested in the myeloid panel.
2. It is stated that patient 3 died from MDS, but it would be helpful if authors could add the cause of death.
3. It would be very helpful if authors could add more details about the morphologic abnormalities post-CAR-T therapy to Table 4.
Author Response
We appreciate very much the reviewer's positive notes. Below are our specific responses to the comments.
Comment 1: the list of the mutations that were examined has been added as Supplementary Material.
Comment 2: the cause of death (infection) has been specified
Comment 3: we added specific morphological MDS features found in the patients in section 3.1
Reviewer 3 Report
The authors report 5 cases of MDS developing aftzer BCMA-targeted CAR-T therapy in five heavily pretreated patients. This is a very important topic, needs to be fully explored, as although CRA-T was given to a heavily pretreated population with not soo long extimated survival, but still the developing MDS may have shirtened the survial of patients. Knwoing this beforhand other therapeutic choice (bispecific AB?) may have been choosen. This this report contains vital information to uinderstand the saftey and short term side effects of BCMA CAR-T therapy.
There is a recent important publication from DFC (Samur MK et al. in Blood April 6. 141(14):1724) anaylsisng the effect of high dose melphalan on the mutational burden of myeloma cells.
The Table 1. the authors summarize the patient's chaarcteristics. Prior treatments are listed, and in a seperate line prior ASCT is also listed. IT would be essential to know the ASCT conditioning regimens in the patients, as it may have a strong influence on later dysplasia, please add this information to table 1.
The data presented by the authors is very important, that they hed pre CAR-T samples and did a retrospective anaylsis for the presence of MDS related mutation. Their data of showing these mutations in lower frequency before CAR-T postulates, taht CAR-T alone may not promote muztations, but can alter the survival benefit of different clones, amking the MDS clone worse, causing clinical symptoms.
I think this is a nice, and important paper, recommend th publication afetr minor changes.
English language used is fine.
Author Response
We appreciate the reviewer's positive notes.
As requested, information on the ASCT conditioning has been added to Table 1